# Enzyme-Mediated Exponential Glucose Release: A Model-Based Strategy for Continuous Defined Fed-Batch in Small-Scale Cultivations

**DOI:** 10.3390/bioengineering11020107

**Published:** 2024-01-24

**Authors:** Annina Kemmer, Linda Cai, Stefan Born, M. Nicolas Cruz Bournazou, Peter Neubauer

**Affiliations:** Institute of Biotechnology, Chair of Bioprocess Engineering, Technische Universität Berlin, 13355 Berlin, Germany; kemmer@tu-berlin.de (A.K.); l.cai@tu-berlin.de (L.C.); stefan.born@tu-berlin.de (S.B.); mariano.n.cruzbournazou@tu-berlin.de (M.N.C.B.)

**Keywords:** glucose release, high-throughput bioprocess development, exponential fed-batch, scale-down, glucoamylase

## Abstract

Miniaturized cultivation systems offer the potential to enhance experimental throughput in bioprocess development. However, they usually lack the miniaturized pumps necessary for fed-batch mode, which is commonly employed in industrial bioprocesses. An alternative are enzyme-mediated glucose release systems from starch-derived polymers, facilitating continuous glucose supply. Nevertheless, while the glucose release, and thus the feed rate, is controlled by the enzyme concentration, it also strongly depends on the type of starch derivative, and the culture conditions as well as pH and temperature. So far it was not possible to implement controlled feeding strategies (e.g., exponential feeding). In this context, we propose a model-based approach to achieve precise control over enzyme-mediated glucose release in cultivations. To this aim, an existing mathematical model was integrated into a computational framework to calculate setpoints for enzyme additions. We demonstrate the ability of the tool to maintain different pre-defined exponential growth rates during *Escherichia coli* cultivations in parallel mini-bioreactors integrated into a robotic facility. Although in this case study, the intermittent additions of enzyme and dextrin were performed by a liquid handler, the approach is adaptable to manual applications. Thus, we present a straightforward and robust approach for implementing defined continuous fed-batch processes in small-scale systems, where continuous feeding was only possible with low accuracy or high technical efforts until now.

## 1. Introduction

In aerobic biotechnology processes, carbon-limited fed-batch is the preferred cultivation mode to prevent substrate accumulation, by-product formation, or engineering limitations such as oxygen transfer rate or excessive heat production. However, during the very early steps of process development, batch mode cultivations are often preferred to allow for big numbers of parallel cultivations, e.g., for clone screening. This poses significant challenges since the performance of the cells strongly depends on their specific growth rate. Consequently, unexpected outcomes arise during scale-up when the best candidates from the batch screenings are transferred to the fed-batch conditions. To avoid these difficulties, the fed-batch technology should already be applied during the early screening phase [1,2,3].

Several approaches for fed-batch strategies in miniaturized systems have been developed in recent years—see Teworte et al. [4] for a review on feeding methods in small-scale systems. However, the challenge of developing an accurate and simple continuous feeding technology remains. In miniaturized systems, the feeding of the carbon source is either not possible at all or discontinuous—either by droplet formation at low pump rates or by bolus feeding through the pipetting needles of a liquid-handling robot. These oscillations of substrate availability have significant impacts on the culture compared to continuous feeding. It is well-known that short periods of substrate excess followed by starvation have an influence on growth and product formation [2,5]. While controlled oscillations can be a valuable tool to mimic the concentration gradients present in industrial-scale bioreactors and their impact on microbial physiology [5,6], the effects of discontinuous additions of the carbon source limit the comparability of small-scale cultivations to ideally mixed systems, e.g., the benchtop-scale scale.

Methods for continuous feed include miniaturized pumps [7,8], glucose release by diffusion from a polymer matrix [9,10,11,12] or enzyme-mediated glucose release from an insoluble [13] or soluble polysaccharide [14]. Miniaturized pumps offer the advantage of a close similarity to benchtop bioreactors, but are difficult to implement, maintenance intensive and tend to become clogged. In the case of glucose release from a polymer matrix, controlling the feed rate is challenging, and its implementation in stirred systems is not straightforward. This is because gels or beads containing the substrate need to be added [9,10]. Finally, enzyme-mediated glucose release systems [13,14] that operate by releasing glucose monomers from starch or dextrins through hydrolysis of glycosidic bonds by glycolytic enzymes such as glucoamylase are the most robust and suited solution for miniaturized cultivation systems. This technology, called Enbase^®^, has been successfully applied for expression of different difficult-to-express proteins in *Escherichia coli* in shake flasks and microwell plates, such as single-domain antibodies [15] or ribonuclease inhibitors [16], as well as during online-adaptive experiments for growth characterization of *E. coli* in mini-bioreactors (MBRs) [17] and microtiter plate cultivations [18].

In applications with an enzyme-mediated substrate delivery, the feed rate depends on the concentrations of the reaction components. As all components are in a liquid state, continuous glucose feeding with adjustable rates can be achieved by bolus additions of either the biocatalyst or the polymeric substrate [18]. This method thus offers the potential of being handled manually by technicians in the lab or automatically using liquid handlers. Figure 1 schematically shows the cultivation course with respect to glucose availability during a pumped continuous glucose feed in a benchtop-scale bioreactor (a), discontinuous, bolus-based feed (b), as well as continuous enzyme-mediated feed (c). Here, the latter approach potentially enables cultivation in the small scale under conditions similar to continuous feed.

However, it is difficult to plan an accurate feeding profile since the hydrolysis reaction shows a non-linear, highly complex behavior, which depends on various environmental conditions. The concentrations of enzyme, polymer substrate and glucose, the composition of the polysaccharide [19], the pH [20,21] and the temperature [22,23] influence the glucose release rate. Other impacting factors include the polysaccharide chain length [24], high viscosity of concentrated starch solutions [25], branching [26], competition between different types of glycosidic bonds [27], different binding modes of the enzyme to the substrate [28] and enzyme inactivation [29]. Efforts to describe these dependencies with the help of mathematical models exist. While data-driven models exist for starch, such as examples found in [30,31], the more prevalent approach has been the proposal of mechanistic models employing Michaelis–Menten type kinetics [24,25,26,32]. However, these models have been developed in the context of saccharification processes in the food industry. Specifically, they are designed for the production of high-glucose syrups under conditions that are not favorable for microbial cultivations, such as acidic pH or temperatures exceeding 40 °C. Thus, they are not directly applicable for enzyme-mediated glucose release in microbial cultivations.

In this study, we present an approach to predict and plan the glucose release to achieve continuous fed-batch conditions with an adapted feed rate control in small-scale cultivations. After the evaluation of different mechanistic starch hydrolysis models presented in literature, the model by Polakovič and Bryjak [32] was selected and adapted under conditions relevant for microbial growth. The model was embedded into a modeling and bioprocess control framework [33], which allows the prediction of the glucose release and the progression of the microbial cultivation. The additions of enzyme needed to achieve specific glucose release rates were calculated and automatically realized during parallel MBR cultivations by a liquid handler. We show that, using this approach, substrate-limited exponential growth without oscillations in the substrate availability is achieved. 

## 2. Materials and Methods

### 2.1. General Experimental Set-Up

All experiments were performed in the mL-scale in a high-throughput platform [34]. The cultivation platform consists of a Tecan Freedom Evo 200 liquid handling site (LHS) (Tecan Group Ltd., Männedorf, Switzerland) hosting a bioREACTOR^®^ 48 system (2mag AG, Munich, Germany) for up to 48 experiments in stirred MBRs. The platform allows for sophisticated process handling by the automated additions of liquids, sampling, as well as temperature control, aeration and agitation by magnetic stirrers. Setpoints for enzyme, glucose and dextrin additions, pH and temperature control, as well as all measurement data are stored in a central database. All experiments were performed at a temperature of 30 °C. Evaporation at the cultivation conditions was determined to be approx. 30 µL h^−1^. In the case that more volume was lost due to evaporation than was added due to feed and pH control, evaporation control was executed by the additions of deionized water by the LHS. Otherwise, no evaporation control was performed to avoid dilution of the medium.

If not stated differently, the experiments with and without cells were conducted in mineral salt medium (MSM) with varying initial concentrations of glucose and dextrin. All chemicals were purchased from Roth, VWR or Merck, if not stated otherwise. For the MSM medium, the following solutions were prepared and sterilized separately: basic salt solution (autoclaved) containing (final medium concentration; the medium was prepared as 5× concentrate) 12.6 g L^−1^ K_2_HPO_4_, 3.6 g L^−1^ NaH_2_PO_4_ × 1H_2_O, 2 g L^−1^ Na_2_SO_4_, 2.468 g L^−1^ (NH_4_)_2_SO_4_, 0.5 g L^−1^ NH_4_Cl, 1 g L^−1^ (NH_4_)_2_-H-citrat; dextrin concentrate (autoclaved) containing 150 g L^−1^ maltodextrin (dextrose equivalent (DE) ≤ 5, EnPump™, EnPresso GmbH, Berlin, Germany); magnesium sulfate concentrate (autoclaved) containing 246.48 g L^−1^ MgSO_4_ × 7H_2_O; thiamine concentrate (sterile filtered) containing 50 g L^−1^ thiamine hydrochloride; trace element solution (sterile filtered) containing (final medium concentration; the medium was prepared as 5× concentrate) 0.5 g L^−1^ CaCl_2_ × 2H_2_O, 0.18 g L^−1^ ZnSO_4_ × 7H_2_O, 0.1 g L^−1^ MnSO_4_ × H_2_O, 20.1 g L^−1^ Na_2_-EDTA, 16.7 g L^−1^ FeCl_3_ × 6H_2_O, 0.16 g L^−1^ CuSO_4_ × 5H_2_O, 0.725 g L^−1^ Ni(NO_3_)_2_ × 6H_2_O; and D-glucose concentrate (autoclaved) containing 600 g L^−1^ D-glucose (C_6_H_12_O_6_). For 1 L final medium, 200 mL basic salt solution, 2 mL magnesium sulfate concentrate, 2 mL thiamine concentrate, 2 mL trace element solution, appropriate volumes of dextrin and glucose concentrate and deionized water were combined to reach desired dextrin and glucose concentrations and the target end volume (see Section 2.2 and Section 2.3 for target concentrations and end volume). The pH was adjusted to pH 7.0 using 7 M ammonia. The glucose release was achieved by the addition of a glucoamylase stock solution (3000 U L^−1^, ReagentA, EnPresso GmbH, Berlin, Germany).

Sample plates were prepared by adding 15 µL of 2 M NaOH_(aq)_ to each well of 96-well V-bottom deep well plates (Corning Inc., New York, NY, USA). The plates were dried at 80 °C for at least 24 h. Samples of 200 µL were taken automatically by the pipetting tips of the Tecan LHS, pipetted into the chilled sample plates and mixed to dissolve the dried sodium hydroxide. This results in an increase in the pH, reaching up to pH 13, which stops the glucose release by inactivating the enzyme and inhibits cell activity. Immediately after, the samples of the cell-free release experiments and the microbial fed-batch cultivations were analyzed as described below. 

### 2.2. Cell-Free Experiments for Selection of the Glucose Release Model

Under sterile conditions, each MBR was filled with 11 mL of the MSM at pH 7.0 containing different initial concentrations of dextrin and glucose (Table 1). The MBR block with 24 experiments was aerated with 0.4 L_n_ min^−1^ (mass flow at 0 °C, 1.013 bar) pressurized air, and agitation was set to 1600 rpm. Samples were taken every 1 to 2 h before and every 2 to 4 h after addition of glucose or dextrin into the sample plates. The glucose concentration was measured directly in the sample using a Cedex Bio HT Analyzer (Roche Holding AG, Basel, Switzerland), which allows for automated analysis of various substrates and metabolites in aqueous solutions. The kinetic experiments were started by enzyme addition to each MBR by the pipetting needles of the Tecan LHS. After approx. 6 h, dextrin and glucose pulses were added to some experiments to evaluate the possibility of substrate and product inhibition. A detailed overview of the experimental conditions is given in the Appendix A.

### 2.3. Enzymatic Fed-Batch Experiments

A total of 39 MBR cultivations were performed in two experimental runs with *Escherichia coli* BL21(DE3), carrying the plasmid pET28-NMBL-eGFP-TEVrec-(*V*_2_*Y*)_15_-His with Kanamycin resistance, expressing a recombinant fusion protein of Elastin-like polypeptide and eGFP, under control of the T7-RNA polymerase, which is induced by isopropyl-β-D-thiogalactopyranosid (IPTG) [35,36]. For a first preculture, 10 mL LB medium containing 16 g L^−1^ tryptone, 10 g L^−1^ yeast extract and 5 g L^−1^ NaCl was supplemented with 10 µL Kanamycin stock (50 mg mL^−1^) and inoculated with 1 mL cryostock (25% glycerol). The preculture was cultivated in a 125 mL UltraYield^®^ Flask sealed with an AirOtop^®^ enhanced flask seal (both from Thomson Instrument Company, Oceanside, CA, USA) for 5 h at 37 °C and 220 rpm in an orbital shaker (50 mm amplitude) (LT-X, Adolf Kühner AG, Birsfelden, Switzerland). Second precultures were performed with 25 mL of EnPresso^®^ B fed-batch medium for microbial cultivations with 6 U L^−1^ Reagent A (both from Enpresso GmbH, Berlin, Germany), 2.5 µL antifoam (PPG2000) and 25 µL kanamycin stock (50 mg mL^−1^) in 250 mL UltraYield flasks sealed with an AirOtop membrane. These cultures were inoculated with the first preculture to an optical density at 600 nm (OD_600_) of 0.25 and cultivated at 37 °C and 220 rpm for 14 h. A conversion factor of OD_600_ to biomass of 0.37 was previously determined and applied throughout the experiments.

Each MBR was filled with MSM (see Section 2.1) and the second preculture was added automatically by the liquid handling needles of the Tecan LHS to an initial OD_600_ of 0.25 in a total of 10 mL medium. An overview of the experimental conditions can be found in Table 2. The setpoints for the feed rates were chosen with regards to a maximal growth rate of the strain *μ*_max_~0.7 h^−1^, determined in preliminary experiments. All experiments were performed in triplicate. The MBR block was aerated with 5 L_n_ min^−1^ pressurized air per 24 MBRs, and agitation was set to 2800 rpm. The agitation was higher than in the cell-free glucose release experiments (see Section 2.2) to avoid oxygen limitation. The cultivations were controlled at the target pH of 7.0 +/− 0.2 by addition of 7 M ammonia or 3 M phosphoric acid, respectively.

A first cultivation run (data set 2) was performed including enzymatic feed with a lower exponential feed rate *μ*_set_ = 0.18 h^−1^ (25% of *μ*_max_ 0.7 h^−1^) during the initial feeding phase and *μ*_set_ = 0.09 h^−1^ (12.5% of *μ*_max_) after induction of recombinant protein production, as well as a slightly higher exponential feed rate *μ*_set_ = 0.21 h^−1^ (30% of *μ*_max_) during the initial feeding phase and *μ*_set_ = 0.11 h^−1^ (15% of *μ*_max_) after induction. Additionally, experiments with only two enzyme additions but a similar target release were performed. Two types of controls were included: (1) cultivations with bolus feed with the same feed rate as during the enzymatic feed experiments and (2) cell-free release experiments to verify that the previously fitted glucose release model is valid. After the batch phase of approx. 9 h, the “control pulse” cultivations were supplied with bolus additions of a 377.3 mg L^−1^ glucose feed by the robotic pipetting tips [34]. For the remaining MBRs, the glucose release, and thus the glucose feed, was initiated and controlled by repeated enzyme additions by the liquid handler. Additionally, 40 µL of the dextrin stock solution were added every 10 min to keep the dextrin concentration high. The frequency of 10 min was due to the time needed to regularly perform culture-handling procedures, such as pH control and sampling. Production of the recombinant protein was induced 6 h after fed-batch start by addition of 0.5 M IPTG stock to a concentration of 0.2 mM.

A second cultivation run (data set 3) was performed similarly but with a wider range of exponential feeding rates (*μ*_set_ = 0.14/0.21/0.25 h^−1^, corresponding to 20%/30%/35% of *μ*_max_, before induction, resp. *μ*_set_ = 0.07/0.11/0.12 h^−1^, corresponding to 10%/15%/17.5% of *μ*_max_, after induction). Furthermore, the glucose release in the cell-free control experiments was initiated with the start of the microbial batch phase to allow for estimation of the parameters of the glucose release model until the fed-batch start for the enzymatic feed experiments.

In the MBRs, the dissolved oxygen tension (DOT) and pH were measured online every 30 to 60 sec by fluorescence sensors (PreSens Precision Sensing GmbH, Regensburg, Germany). At-line samples were taken every 1 to 2 h, starting approximately 1 h before the batch end. The OD_600_ was measured in a Synergy MX microwell plate reader (BioTek Instruments GmbH, Bad Friedrichshall, Germany) after sample dilution with 0.9% NaCl solution. The cells were separated from the supernatant by centrifugation at 15,000× *g* for 10 min, and the concentration of glucose, acetate and magnesium were determined in the supernatant using the Cedex Bio HT Analyzer. An additional estimate of the biomass concentration $X_{NH_{3}}$ was obtained from the added ammonia [33]. The amount of the recombinant product was obtained from fluorescence measurements based on a previously determined conversion factor.

### 2.4. Modeling and Simulation

The model-based adaptive input design was implemented using an in-house developed Python-based framework for simulation [33] with symbolic differentiation [37], and parameter estimation using lmfit [38] and pygmo [39]. For lmfit the method “least_squares”, and for pygmo the algorithm “differential evolution” (“de1220”) with a population size of 20 and generation number of 200 was used. Samplings and inputs, namely the addition of dextrin, glucose, enzyme and pH control reagents acid and base, are given as boluses and therefore represent discontinuous actions. After each of these actions, the simulation is stopped, the volume-dependent states are recalculated, and the simulation is restarted. Please refer to the section “Calculation of dilution by bolus additions” in the Appendix A for equations and explanation. Further details on the simulation and parameter fitting procedure are given in Kemmer et al. [33].

## 3. Results

### 3.1. Selection and Parameter Fitting of the Glucose Release Model

The kinetics of dextrin hydrolysis to glucose molecules by glucoamylase were investigated in 24 experiments under the influence of various levels of the design variables glucose, dextrin, and enzyme concentration (data set 1). To understand the effects of substrate and product inhibition, the influence of dextrin and glucose additions 6 h after initiation of glucose release was evaluated. As the release reaction is dependent on the pH and the temperature, the experiments were controlled at 30 °C and pH 7.0, which are commonly used for microbial cultivations.

The glucose concentration was measured after stopping the release reaction. All experiments show an initially fast glucose release, slowing down with the progression of the reaction. Due to the conversion of anhydroglucose in starch into free glucose (conversion factor of 1.111) [40], the applied dextrin concentrations of 15.0 to 37.5 g L^−1^ yield ~16.6 to 41.7 g L^−1^ of released glucose. However, after 30 h reaction time at close-to-zero reaction rates, only just over half of the initial dextrin concentration was hydrolyzed (Figure 2).

In these experiments, the enzyme-mediated glucose release is achieved by the cleavage of D-glucose molecules from dextrin by the exo-acting enzyme glucoamylase (1,4-α-D-glucan glucohydrolase, EC 3.2.1.3) from *Aspergillus niger*. The enzyme catalyzes the hydrolysis of α-1-4 and α-1-6 glycosidic bonds from the non-reducing ends of starch and related poly- and oligosaccharides [27,41]. 

Data of 24 release experiments were used simultaneously to fit the parameters of different mechanistic glucose release models presented in literature [24,25,26,32] (Table 3). The models center around Michaelis–Menten type release kinetics, including inhibition by the product glucose and the substrate dextrin, differences in substrate types, as well as enzyme inactivation.

During incubation without dextrin, the glucoamylase was found to be stable in preliminary experiments. Thus, no degradation of the enzyme stock solution was assumed. Our model-fitting approaches confirmed previous suppositions [26,27,32] that the glucose release cannot be described using simple Michaelis–Menten type kinetics, or modifications of these kinetics for product inhibition or enzyme deactivation. Our data at pH 7 and 30 °C suggest that a part of the substrate was not susceptible to the enzymatic hydrolysis or at a much slower rate. This is in agreement with the findings of Polakovič and Bryjak [32] who distinguish two fractions of starch, with one fraction much less susceptible to enzymatic hydrolysis. However, in contrast, including inhibition terms did not improve the model fit and was thus omitted. Based on the model considering two substrate fractions with potentially different release rates as proposed by Polakovič and Bryjak [32], we have an ODE system for the variables product glucose P [g L^−1^], susceptible substrate $S_{S}$ [g L^−1^] and a resistant substrate $S_{R}$ [g L^−1^], enzyme E [U L^−1^] and volume V [L]:(1)$$\frac{dP}{dt}=1.111\left(r_{S}+r_{R}\right)-\frac{\frac{dV}{dt}}{V}P$$
(2)$$\frac{dS_{S}}{dt}=-r_{S}-\frac{\frac{dV}{dt}}{V}S_{S}$$
(3)$$\frac{dS_{R}}{dt}=-r_{R}-\frac{\frac{dV}{dt}}{V}S_{R}$$
(4)$$\frac{dV}{dt}=-r_{evap}$$
(5)$$\frac{dE}{dt}=\frac{-\frac{dV}{dt}}{V}E$$

The release rates $r_{S}$ [g (Lh)^−1^] for susceptible substrate and $r_{R}$ [g (Lh)^−1^] for resistant substrate are calculated from the enzyme concentration E, the catalytical constants $k_{S}$ [g (Uh)^−1^] and $k_{R}$ [g (Uh)^−1^], and the Michaelis constant K [g L^−1^]. Only one K-value is included as it was found that the model could not discriminate different affinities of the enzyme towards the two substrate types [32]:(6)$$r_{S}=\frac{k_{S}ES_{S}}{S_{S}+S_{R}+K}$$
(7)$$r_{R}=\frac{k_{R}ES_{R}}{S_{S}+S_{R}+K}$$

We will be able to determine only the total concentration S of both substrates: (8)$$S=S_{R}+S_{S}$$
which yields:(9)$$\frac{dS}{dt}=-r_{S}-r_{R}-\frac{\frac{dV}{dt}}{V}S$$

To rewrite the differential equation with variables P, S, E and V, we introduce a new variable, namely the proportion of $S_{S}$ to S:(10)$$W_{S}=\frac{S_{S}}{S}$$

Thus, we can substitute $S_{S}=W_{S}S$ and $S_{R}=(1-W_{S})S$ in all occurrences. From (6) and (7) in particular, we obtain the rates $r_{S}$ and $r_{R}$ as function of S and $W_{S}$. The dynamics of $W_{S}$ ensues from the quotient rule:(11)$$\frac{dW_{S}}{dt}=\frac{\frac{dS_{S}}{dt}S-\frac{dS}{dt\;}S_{S}}{S^{2}}=\frac{-r_{S}S+S_{S}\left(r_{S}+r_{R}\right)}{S^{2}}$$

Only one K-value is included as it was found that the model could not discriminate different affinities of the enzyme towards the two substrate types [32]. The determined model parameters are shown in Table 4. Presumably, due to the comparatively high dextrin concentrations in the experiments the Michaelis constant K could not be fitted but tended to be in the low mg L^−1^ range. As low K-values were also reported in literature [28,32,42], the K-value was fixed at 0.001 g L^−1^. 

### 3.2. Application of Enzyme-Mediated Glucose Release in Microbial Fed-Batch Cultivations

For the enzymatic glucose feed experiments, the enzyme feed is determined by iterative comparison of the target glucose feed rate and the current glucose release rate. Enzyme additions were planned and executed by the liquid handler to control the glucose release rate and thus the growth at the setpoint (see Table 2). Technical constraints of the liquid handling station limit the additions so that they are performed at fixed times $t_{i}$ on a temporal grid, $∆t=t_{i+1}-t_{i}$. The target glucose release rate is calculated as the difference between the current glucose concentration $P\left(t_{i}\right)$, the target glucose concentration at a future timepoint ${P(t_{i+1})}_{target}$ and the time difference from the current time to this future timepoint ∆t. The enzymatic feed was calculated under the simplifying assumption that for the feeding intervals, difference quotients approximate the derivatives sufficiently well. The intended average release rate in the interval $I_{i}=[t_{i},\;t_{i+1}]$ is: (12)$${r_{target}=\frac{dP}{dt}}_{target}=\frac{{P(t_{i+1})}_{target}-P\left(t_{i}\right)}{∆t}$$

The necessary enzyme addition at time $t_{i}$ to achieve $r_{target}$ was determined from Equations (1), (6) and (7):(13)$$E_{add,i}=E_{target}-E=\frac{\left({\frac{dP}{dt}}_{target}-\frac{dP}{dt}\left(t_{i}\right)\right)S+K}{1.111(k_{S}S_{S}+k_{R}S_{R})},\;\mathrm{for}\;\frac{dP}{dt}(t_{i})\leq {\frac{dP}{dt}}_{target},\;\mathrm{else}0$$

As S≫K, deviations of $r_{target}$, which occur in the interval $I_{i}$ due to the above used approximation, are controlled by changes of $W_{S}$. Those are negligible given that $\frac{r_{S}}{S∆t}\ll 1$ and $\frac{r_{R}}{S∆t}\ll 1$. In the next feeding interval, again the exact integration of the equations is used so that errors do not accumulate. For longer feeding intervals or conditions with low error tolerance this procedure would be inadequate. Instead, a shooting method minimizing ${\left|P{\left(t_{i}\right)}_{integrated}-P{\left(t_{i}\right)}_{target}\right|}^{2}$ with respect to $E_{add,i}$ could be used. In this case, an initial value could be provided by the above approximation.

The results from the kinetic experiments were validated in parallel high-throughput MBR cultivations with enzymatic glucose feed. A previously described microbial growth model [33] was extended with the mechanistic model for the enzyme-mediated glucose release and was fitted to the validation experiments with enzymatic glucose feed. 

Therefore, initially, 24 experiments were performed (data set 2), including control experiments with pulsed feed as well as cell-free release experiments. In the cell-free controls, ~11–15 g L^−1^ glucose was released in the 10 h between initial enzyme addition and end of the experiment. In the microbial cultivations, after the batch phase of 9–10 h, a biomass concentration of approx. 3 g L^−1^ was reached. Glucose release was initiated in the cultivations with enzymatic glucose feed by enzyme addition. Throughout the fed-batch phase, more enzyme was added to control the glucose release at the desired setpoint. Additionally, dextrin was added to maintain a high substrate concentration for the catalytic reaction. Depending on the cultivation conditions, the cells reached a biomass concentration of 6–7 g L^−1^ until the induction of protein production at 6 h of feeding. As described in Section 2, the feed rates were calculated to control the cell growth at different specific growth rates (*μ*_set_) in the exponential preinduction feed phase. During the induction phase of 4 h, approx. 1.2–2 g L^−1^ ELP-GFP were produced. During the preinduction feeding phase, the DOT profile shows a continuous decrease, indicating that a continuous substrate release, and thus consumption, with increasing rate is achieved. Conversely, in the bolus-fed control (Figure 3c), oscillations in the oxygen signal can be observed as cells switch between excess and starvation in between the glucose additions. The amount of glucose that is added is similar to the released glucose in the enzyme-mediated glucose fed cultivations, but the achieved amount of biomass at the end of the fed-batch phase is, as expected for pulse-based feed scenarios, significantly smaller. While the bolus-fed experiments qualitatively show very well the difference in substrate input in comparison to the cultivations with continuous enzyme-mediated feed, the influence for the lower biomass in the bolus-fed experiments cannot be quantified yet. Thus, these experiments are not included in the subsequent model fitting.

First, the parameters of glucose release model, K, $k_{S}$, $k_{R}$ and $W_{S,0}$, were reassessed based on the glucose measurements of the cell-free experiments. The glucose release was slightly overestimated when using the parameters obtained in the model calibration experiments. As shown in Figure 3a, a good fit of the glucose can be achieved with a lower catalytical constant for the susceptible substrate $k_{s}=0.076$ g (UL)^−1^. The lower catalytical constant indicates that less glucose was released than originally planned. This also explains the deviation of the actual growth rates from the setpoints. The simulated concentrations of the susceptible and resistant dextrin fractions show an accumulation of the resistant dextrin with increased experimental time. This is presumably due to the extremely low hydrolysis of the resistant substrate—$k_{S}$ is 60 times larger than $k_{R}$—and simultaneous addition of dextrin stock during the feeding phase to maintain a sufficient concentration of susceptible substrate. In a next step, with fixed glucose release parameters, the growth-related model parameters were estimated using the data of the experiments with enzyme-mediated glucose feed (see Appendix A). The model can describe the course of the biomass, product and glucose data with sufficient accuracy (Figure 3b) and shows the trends for the DOT profile.

Due to the lower catalytical constant in the first cultivation run (data set 2) in comparison to the initial model fitting experiments (data set 1), lower growth rates than expected were observed. To avoid such deviations, we performed a second cultivation run (data set 3), where the glucose hydrolysis in cell-free controls was started with the beginning of the microbial batch phase. Until the start of the microbial fed-batch phase after ~10 h, we were able to obtain the model parameters valid under the exact experimental conditions. Here, the parameters for the glucose release model of data set 2 were confirmed. The obtained parameters were applied to calculate setpoints for enzyme additions to realize target glucose release rates. 

Three different growth rates could be achieved (see Figure 4 for examples). A comparison between the setpoints for the exponential growth rate and the measured growth rate is shown in Table 5. Before induction, the setpoints could be achieved with a maximal deviation of ~15%, whereas after induction, growth was rather uniform at a growth rate of ~0.06 h^−1^ regardless of the setpoints. Thus, the growth is limited by the metabolic load of the cell rather than by the glucose feed.

## 4. Discussion

A method to enable continuous glucose feed in small-scale microbial cultivation systems was developed using mechanistic modeling. Glucose cleavage was catalyzed by the amylolytic enzyme glucoamylase.

Initially, a suitable model was chosen from the approaches presented in the literature. These models were initially developed within the context of saccharification processes in the food industry, operating under conditions that are not suitable for microbial growth. The objective of this study was the selection of a model under conditions favorable for microbial cultivations and beneficial for the production of recombinant proteins, specifically, at 30 °C and a pH of 7.0. Data from cell-free glucose release experiments were successfully fitted using a Michaelis–Menten kinetic extended by the assumption of two different substrates, a fraction that is more susceptible to hydrolysis and a resistant one [32]. Product inhibition [28,32] and substrate inhibition [25,26] do not seem to play a role in our application. The estimated parameter values for the catalytical constants for the susceptible and resistant dextrin, $k_{S}$ and $k_{R}$, are in a similar range than reported in Polakovič and Bryjak [32], while the initial relative amount of susceptible substrate, $W_{S,0}$= 0.46%, was lower than the ~77% reported in previous publications [32,43]. This is reasonable as the hydrolysis reaction is pH dependent [21] and the experimental conditions in this study are far from the optimal conditions of the enzyme glucoamylase, which are a pH of 4.5 to 5 and temperature of 60 °C [44,45]. Further analysis of the two fractions would be necessary for a quantitative explanation of the behavior. However, as Polakovič and Bryjak [32] use soluble starch with a DE of 0.4 in comparison to the maltodextrin with a DE of ≤5 used in this study, a direct comparison is not possible and not the aim of this study. 

The glucose-release model was combined with a macro-kinetic model [46] describing the growth of *E. coli* in high-throughput enzyme-mediated glucose feed experiments [33]. The combined model was applied in two experimental runs to calculate necessary glucoamylase additions to achieve setpoints for glucose feed during cultivation of *E. coli* producing a recombinant protein. In both runs, cell-free controls were performed to assess the validity of the glucose release model. Data of cell-free control experiments in the first cultivation run (data set 2) showed that $k_{s}$ estimated to be 0.0756 g (Uh)^−1^ and is thus smaller than the 0.134 g (Uh)^−1^ estimated in the previous model identification experiments. Consequently, the attained glucose release rates, and thus the microbial growth rates, were lower than the setpoints (Table 2). The reasons for this might be the higher agitation and aeration, which were necessary to avoid oxygen limitation in the microbial cultivations, as those can lead to inactivation of the enzyme and thus a lower activity [47]. Additionally, a negative effect on the reaction rate has been observed due to limitations in the mass transfer at higher dextrin concentrations [26], which is the case in the enzymatic fed-batch experiments. 

In the second cultivation run (data set 3), the glucose release in the control experiments was assessed during the microbial batch phase. Here, the model parameters that were obtained during the previous cultivation run were confirmed. The setpoints for the growth rates could be achieved with an acceptable error of 15%.

The bolus-fed cultivations show a significantly lower biomass growth than the continuously fed cultures despite comparable glucose supply. This is the typical well-described response of many microbial cells to the oscillating feast vs. famine availability of glucose (feed zone effect) [5]. Our results thus suggest that the influence of oscillations can be studied in these small-scale systems. This confirms the potential use of miniaturized cultivations to study scale effects [6], and would be valuable to include in further studies.

The main aim of applying an enzyme-mediated glucose release in this study was to enable continuous glucose feed in small-scale cultivation systems, i.e., avoiding oscillations in the glucose availability. As *E. coli* has a fast glucose transport system, the cells react instantaneously to changes in the availability of this carbon source, which in turn leads to abrupt changes in the oxygen concentration. This is reflected in the peaks in the oxygen signal of bolus-fed cultivations (see Figure 3c), which are characteristic for the repeated switch between maximum growth after the pulse has been given, and starvation upon consumption of the glucose shortly after. In contrast, all cultivations fed by glucose release show a mainly smooth, steadily decreasing course of the oxygen signal (see Figure 3b for an example), indicating that continuous substrate release has been achieved. 

Some challenges of the system remain. Accumulation of resistant dextrin and low release rates set limits in the possible length and setpoints for the release rate. We identified a setpoint for the growth rate *µ*_set_ = 0.197 h^−1^ for 6 h followed by *µ*_set_ = 0.088 h^−1^ for 4 h corresponding to a glucose release of ~22 g L^−1^ from 80 g L^−1^ dextrin as the upper boundary. While dextrin is still present in the medium, large volumes of enzyme would be necessary to (theoretically) maintain the target release rates. Here, the addition of debranching enzymes, cleaving the α-1-6 glycosidic bonds, [48] could improve the release rate. An alternative would be to allow for decreasing substrate release rates at the later stages of the cultivation. Additionally, the release rate cannot be reduced actively. A reduction in the release rate only occurs due to the decrease in the overall available substrate as a result of the hydrolysis as well as due to accumulation of the resistant dextrin. Thus, decreasing setpoints of the release rate might not be achievable. As the release depends on the relation of the susceptible to resistant substrate, the pH and the temperature, we recommend calibration of the model parameters to each lot of the substrate as well as to changes in the cultivation conditions. Here, the user can obtain an approximation of the experimental data required to ensure sufficiently accurate model predictions using model based experimental design methods [49,50,51]. The aim should be to make sure that the uncertainty in the model outputs at the timepoint of the next update is small enough to guarantee a stable feeding profile.

Nevertheless, our results show that continuous glucose feeding with defined release rates is possible in small-scale systems. Previously, Jansen et al. [18] applied enzymatic glucose feed to regulate the growth rate in microscale experiments. While their method presents a straightforward control approach, it does not allow to distinguish between different factors that impact growth, namely a lack of enzyme, lack of susceptible substrate or cellular factors, such as adverse reactions towards induction of recombinant proteins. To our knowledge, our approach is thus the first implementation of enzyme-mediated glucose feed with controlled feed rates based on knowledge about the mechanism of the glucose release reaction.

In this publication, the case study was executed in an automated cultivation facility, and the enzyme and dextrin additions were performed by a liquid handler. However, given that the pH is stable (i.e., by using a buffer system), this approach is transferable to small-scale systems where liquid additions are performed by hand. For this, in the feed calculator of the simulation framework (provided in the Appendix A) the initial conditions of the system and the desired growth setpoints can be adapted to calculate setpoints for enzyme and dextrin additions. For manual applications, we suggest cumulation of additions that are close in time. The simulation framework can subsequently be used to verify whether the release rates that are achieved using this simplified approach are close to the target. 

## 5. Conclusions

We present an approach for defined continuous glucose feeding in small-scale systems. A glucose-release model considering two substrate fractions was adapted to conditions relevant to microbial cultivations. Subsequently, we demonstrated the usage of enzymatic glucose feed in validation experiments with *Escherichia coli*. Different exponential growth rates could be achieved by intermittent additions of enzyme and dextrin. This approach mimics the continuous addition of a glucose solution via pumps, which is the main feeding strategy at industrial scale. Therefore, our approach enables small-scale screening of potential production strains and cultivation settings close to industrial conditions. To facilitate the use of the glucose release model in a broader application spectrum, the influence of the temperature and the pH should be studied and additionally included in the model. Furthermore, future work will tackle the accumulation of the resistant dextrin fraction. Additionally, it would be interesting to study the influence of oscillations in the glucose availability in comparison the continuous glucose feed on the growth and recombinant protein formation in microbial systems.

## Figures and Tables

**Figure 1 bioengineering-11-00107-f001:**
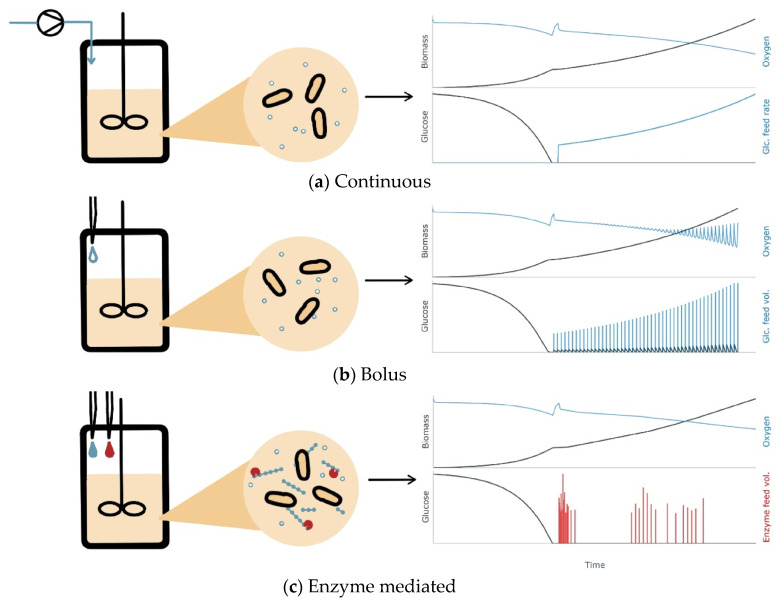
Comparison of (**a**) pumped, (**b**) bolus and (**c**) enzyme-mediated feed in microbial cultivations. In the larger scale, glucose feed is added continuously via pumps while in miniaturized systems discreet pulses are given. Continuous glucose feed in the small scale can be realized by enzyme-catalyzed glucose release from a glucose polymer. The glucose release rate is tuned by additions of enzyme or the polymer either manually using pipettes or by the pipetting needles of a liquid-handling robot.

**Figure 2 bioengineering-11-00107-f002:**
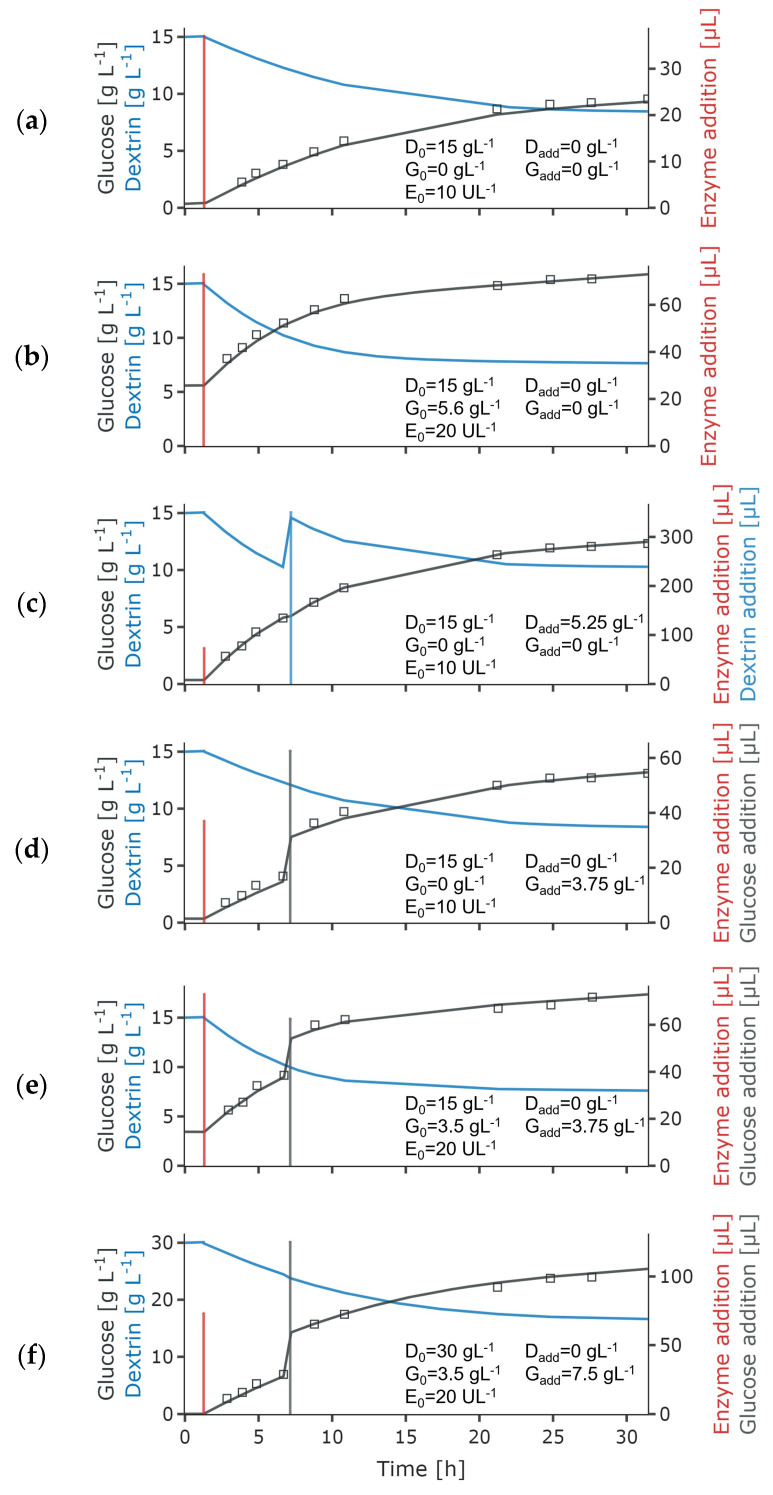
Glucose release experiments under conditions relevant for microbial growth, and model simulation using a Michaelis–Menten type kinetic, considering two types of dextrin [32]. The model parameters were simultaneously fitted to the glucose measurements of 24 release experiments. The subplots (**a**–**f**) show examples under different initial conditions regarding enzyme ($E_{0}$), glucose ($G_{0}$), and dextrin ($D_{0}$) concentration, as well as additions of glucose ($G_{add}$) and dextrin ($D_{add}$), which are given in the subplot. The glucose (measurements—black dots, simulation—black line) and dextrin concentrations (simulation—blue line) are shown, as well as the added volumes of enzyme (red vertical line), glucose (grey vertical line) and dextrin (light blue vertical line).

**Figure 3 bioengineering-11-00107-f003:**
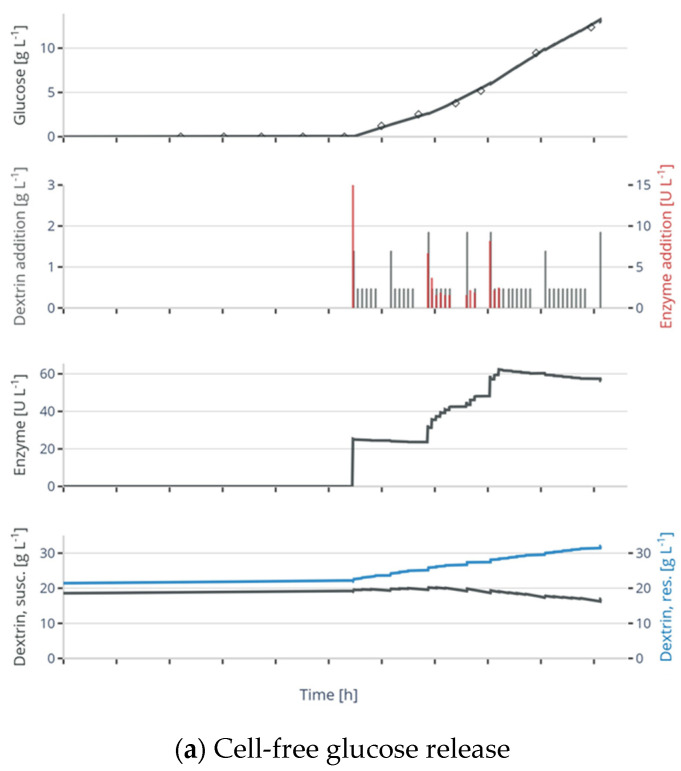
Application of the mechanistic glucose release model to facilitate continuous feeding in *E. coli* cultivations. Glucose is released continuously by enzyme-mediated hydrolysis of dextrin and (**a**) accumulates in cell-free control experiments, or (**b**) is metabolized by the cells of a microbial cultivation. In (**c**) cell growth during a cultivation with bolus additions of glucose feed is shown as comparison. The bars represent the bolus-based additions of dextrin and enzyme, or glucose to the MBRs. The measurements and simulations for the concentrations of biomass and product (for experiments with cells), the dissolved oxygen tension (DOT), the concentration of the susceptible and resistant dextrin, the enzyme concentration are shown, as well as the glucose, which is either (**b**) released or (**c**) added during the feeding. Model simulations are depicted as lines and measurements as symbols or a dotted line (for oxygen). The vertical line indicates the point of induction of the recombinant product formation by IPTG as described in the Section 2.

**Figure 4 bioengineering-11-00107-f004:**
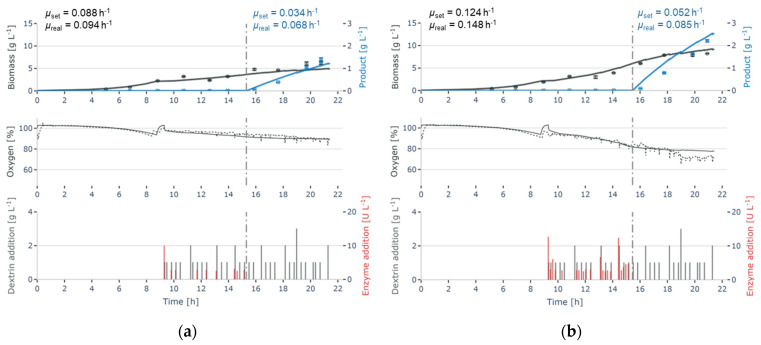
Continuous enzymatic glucose feed of *E. coli* cultivations with two different feeding rates. (**a**) shows the cultivation course with a setpoint for the exponential growth rate *µ*_set_ = 0.088 h^−1^ (before induction), resp. *µ*_set_ = 0.034 h^−1^ (after induction) and (**b**) shows the cultivation course with a higher *µ*_set_ = 0.124 h^−1^ (before induction), resp. *µ*_set_ = 0.052 h^−1^ (after induction). Model simulations are shown as lines, measurements as symbols or a dotted line (DOT) and inputs as bars. In addition to the biomass measurements and simulations, the oxygen tension, as well as the enzyme and dextrin additions are shown.

**Table 1 bioengineering-11-00107-t001:** Design variables of the enzymatic release experiments (data set 1). The initial concentration of dextrin $D_{0}$, glucose $G_{0}$ and enzyme $E_{0}$ were varied. To some experiments, dextrin $D_{add}$ and glucose $G_{add}$ was added after approx. 6h.

Parameter	Description	Unit	Values
$D_{0}$	Initial dextrin concentration	g L^−1^	15, 30
$G_{0}$	Initial glucose concentration	g L^−1^	0, 3.75, 7.5, 15
$E_{0}$	Initial enzyme concentration	U L^−1^	10, 20
$D_{add}$	Dextrin addition	g L^−1^	0, 5.25, 10.5
$G_{add}$	Glucose addition	g L^−1^	0, 3.75, 7.5

**Table 2 bioengineering-11-00107-t002:** Experimental conditions of the two enzymatic fed-batch experiments. Each condition was performed in triplicate. The conditions differed in their feed methods and set growth rates (before and after induction).

Condition	Feed Method	*μ*_set_ [h^−1^] before ind | after ind	$G_{0}$ [g L^−1^]	$D_{0}$ [g L^−1^]	Additions	Cells
**Data set 2**						
1—control pulse	Bolus feed	0.18 | 0.09	5	2	Glucose	Yes
2—control pulse	Bolus feed	0.21 | 0.11	5	2	Glucose	Yes
3—enzymatic feed	Enzymatic release	0.18 | 0.09	5	40	Dextrin, enzyme	Yes
4—enzymatic feed	Enzymatic release	0.21 | 0.11	5	40	Dextrin, enzyme	Yes
5—enzymatic feed	Enzymatic release (two additions)	0.18 | 0.09	5	40	Dextrin, enzyme	Yes
6—enzymatic feed	Enzymatic release (two additions)	0.21 | 0.11	5	40	Dextrin, enzyme	Yes
7—control cell-free	Enzymatic release	0.18 | 0.09	0	40	Dextrin, enzyme	No
8—control cell-free	Enzymatic release	0.21 | 0.11	0	40	Dextrin, enzyme	No
**Data set 3**						
9—enzymatic feed	Enzymatic release	0.14 | 0.07	5	40	Dextrin, enzyme	Yes
10—enzymatic feed	Enzymatic release	0.21 | 0.11	5	40	Dextrin, enzyme	Yes
11—enzymatic feed	Enzymatic release (two additions)	0.25 | 0.12	5	80	Dextrin, enzyme	Yes
12—control cell-free	Enzymatic release	0.14 | 0.07	0	40	Dextrin, enzyme	No
13—control cell-free	Enzymatic release	0.21 | 0.11	0	40	Dextrin, enzyme	No

**Table 3 bioengineering-11-00107-t003:** Residual sum of squared errors for mechanistic models centering around Michaelis–Menten type kinetics (MM) applied to describe glucose release data (see Appendix A for model equations).

Description	Residual Sum of Squared Errors
Simple MM	4.54
Simple MM with product inhibition	3.17
Simple MM with substrate inhibition	4.51
Simple MM with product and substrate inhibition	4.51
MM considering two substrates	1.25
MM considering two substrates with product inhibition	1.18
MM considering two substrates with product and substrate inhibition	1.14

**Table 4 bioengineering-11-00107-t004:** Fitted parameters of the enzymatic glucose release model with standard deviation.

Parameter	Unit	Value	Standard Deviation
K	g L^−1^	0.001	(fixed)
$k_{S}$	g (U h)^−1^	0.134	0.00435 (3.25%)
$k_{R}$	g (U h)^−1^	0.00212	0.000459 (21.68%)
$W_{S,0}$	g g^−1^	0.464	0.0107 (2.31%)

**Table 5 bioengineering-11-00107-t005:** Comparison of setpoints for the growth rate *µ*_set_ and actual reached values *µ*_real_ before and after induction of recombinant protein production.

Before Induction		After Induction	
*µ*_set_ [h^−1^]	*µ*_real_ [h^−1^]	*µ*_set_ [h^−1^]	*µ*_real_ [h^−1^]
0.088	0.093 ± 0.013	0.034	0.066 ± 0.005
0.124	0.142 ± 0.012	0.052	0.063 ± 0.021
0.197	0.167 ± 0.026	0.088	0.051 ± 0.000

## Data Availability

The original contributions presented in the study are publicly available. This data can be found here: https://git.tu-berlin.de/bvt-htbd/public/kemmer_2022_enzymatic-feed.git.

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
