# Peer review of "Enzyme-Mediated Exponential Glucose Release: A Model-Based Strategy for Continuous Defined Fed-Batch in Small-Scale Cultivations"

_bioengineering, 2024, doi:10.3390/bioengineering11020107_

Round 1

Reviewer 1 Report

Comments and Suggestions for Authors

Overall a very interesting paper and highly valuable in the synthetic biology space where screening large numbers of strains under industrial process conditions is important.  The following are some points for the authors to consider:

1) Please clarify for the readers how the reduction in growth rate is achieved at the time of induction.  In lines 286 & 287, the authors state the glucoamylase is stable and therefore the periodic addition of this enzyme is additive and contributes to the exponential glucose release.  Is the reduction of growth rate solely dependent on the accumulation of the "less susceptible" starch/dextrin fraction?  If so, the authors should comment on the lot-to-lot consistency of this substrate.  Do they recommend calibrating each lot of dextrin or starch to determine the "less susceptible" fraction and adjust the model accordingly?

2) I would recommend the authors include the amount of acetate accumulated in the culture medium, comparing the enzymatic glucose release to the bolus addition of glucose.  The data were collected using the Cedex instrument but I do not see it presented.  This would address one of the major issues with the use of glucose as a substrate. 

Comments on the Quality of English Language

Overall the English grammar is fine but the authors should go over the paper one last time to pick up any errors

Author Response

Dear reviewer, 

thank you very much for the thorough revision and the helpful comments. 

Please find attached our responses to the comments of you and the second reviewer. We hope that we have been able to answer the questions adequately.

We have revised the manuscript accordingly.

Best regards,

Annina Kemmer

Reviewer 2 Report

Comments and Suggestions for Authors

Small-scale bioprocess experiments struggle with controlled feeding. Enzyme-based glucose release offers an alternative, but it's hard to control. The authors developed a model-based approach that precisely controls enzyme release, enabling accurate continuous feeding in mini-bioreactors, paving the way for easier small-scale bioprocess development. The methods proposed are very relevant to the industrial strain screening process, and the paper is well written. I have a few comments for the authors to consider to strengthen their manuscript.

Major:

Page 7 lines 297 - 299:

The model presented here is a system of ordinary differential equations (ODEs, not DAEs). However, it lacks clarity in several aspects. Firstly, where is the term for the addition of Enzyme (E) or substrate (S) in the vector field of the ODEs? While Fig. 2 C to F clearly show these feed inputs, their representation within the model remains unclear. Secondly, the derivation of equation 3 requires explanation.

Page 19 equation (8):

While the approximation of derivatives using Euler's method may be acceptable for linear functions (potentially even more accurate with small step sizes), it becomes unsuitable when P is a nonlinear function of time, as is often the case in fermentation. Therefore, it's puzzling why the authors didn't opt for symbolic differentiation. CasADi's advantage lies in its ability to leverage both forward and reverse-mode automatic differentiation on expression graphs for efficient gradient calculation.

Page 20 equation (10):

Again, deriving dE/dt here is not the best practice. Instead, a "shooting method" would be more appropriate. This method allows you to simulate the kinetic model forward and test whether adding a specific amount of enzyme can achieve the desired boundary value (e.g., P_target).

Lastly, the authors should consider providing general guidelines for data collection, specifically the number of data points and replicates needed in a time-series run to build a robust model with low uncertainty. This would be valuable for researchers who might replicate the procedure with different carbon sources and enzymes.

Minor:

In Figure 1, the final sentence ("This is a figure. Schemes follow...") reads like a placeholder instruction. Please revise it to provide relevant information.

Parameter G_add is missing a definition in Table 1.

Author Response

(The authors gave the same response as above.)

Round 2

Reviewer 2 Report

Comments and Suggestions for Authors

The authors have addressed my comments properly. I would recommend the paper to be accepted in its current form.